# Sugar-Sweetened Beverage Consumption and Calcified Atherosclerotic Plaques in the Coronary Arteries: The NHLBI Family Heart Study

**DOI:** 10.3390/nu13061775

**Published:** 2021-05-22

**Authors:** Yash R. Patel, Tasnim F. Imran, R. Curtis Ellison, Steven C. Hunt, John Jeffrey Carr, Gerardo Heiss, Donna K. Arnett, James S. Pankow, J. Michael Gaziano, Luc Djoussé

**Affiliations:** 1Division of Aging, Department of Medicine, Brigham and Women’s Hospital and Harvard Medical School, Boston, MA 02120, USA; tasnimfimran@gmail.com (T.F.I.); jmgaziano@partners.org (J.M.G.); ldjousse@rics.bwh.harvard.edu (L.D.); 2Lifespan Cardiovascular Institute, Warren Alpert Medical School at Brown University, Providence, RI 02903, USA; 3Section of Preventive Medicine & Epidemiology, Boston University, Boston, MA 02118, USA; ellison@bu.edu; 4Department of Genetic Medicine, Weill Cornell Medicine, Doha 84132, Qatar; steve.hunt@utah.edu; 5Cardiovascular Genetics Division, University of Utah School of Medicine, Salt Lake City, UT 84132, USA; 6Department of Radiology, Cardiovascular Medicine and Biomedical Informatics, Vanderbilt University Medical Center Nashville, TN 37232, USA; j.jeffrey.carr@vanderbilt.edu; 7Department of Epidemiology, School of Public Health, The University of North Carolina at Chapel Hill, Chapel Hill, NC 27599, USA; gerardo_heiss@unc.edu; 8Division of Epidemiology, University of Kentucky, Lexington, KY 40536, USA; donna.arnett@uky.edu; 9Division of Epidemiology and Community Health, University of Minnesota, Minneapolis, MN 55455, USA; pankow@epi.umn.edu; 10Massachusetts Veterans Epidemiology and Research Information Center (MAVERIC) and Geriatric Research, Education, and Clinical Research Center (GRECC), Boston Veterans Affairs Healthcare System, Boston, MA 02120, USA

**Keywords:** soda consumption, sugar-sweetened beverages, nutrition, atherosclerosis, coronary calcium

## Abstract

Background: Sugar-sweetened beverage (SSB) intake is associated with higher risk of weight gain, diabetes, hypertension, cardiovascular disease, and cardiovascular mortality. However, the association of SSB with subclinical atherosclerosis in the general population is unknown. Objective: Our primary objective was to investigate the association between SSB intake and prevalence of atherosclerotic plaque in the coronary arteries in The National Heart, Lung, and Blood Institute (NHLBI) Family Heart Study. Methods: We studied 1991 participants of the NHLBI Family Heart Study without known coronary heart disease. Intake of SSB was assessed through a semi-quantitative food frequency questionnaire. Coronary artery calcium (CAC) was measured by cardiac Computed Tomography (CT) and prevalent CAC was defined as an Agatston score ≥100. We used generalized estimating equations to calculate adjusted prevalence ratios of CAC. A sensitivity analysis was also performed at different ranges of cut points for CAC. Results: Mean age and body mass index (BMI) were 55.0 years and 29.5 kg/m^2^, respectively, and 60% were female. In analysis adjusted for age, sex, BMI, smoking, alcohol use, physical activity, energy intake, and field center, higher SSB consumption was not associated with higher prevalence of CAC [prevalence ratio (95% confidence interval) of: 1.0 (reference), 1.36 (0.70–2.63), 1.69 (0.93–3.09), 1.21 (0.69–2.12), 1.05 (0.60–1.84), and 1.58 (0.85–2.94) for SSB consumption of almost never, 1–3/month, 1/week, 2–6/week, 1/day, and ≥2/day, respectively (p for linear trend 0.32)]. In a sensitivity analysis, there was no evidence of association between SSB and prevalent CAC when different CAC cut points of 0, 50, 150, 200, and 300 were used. Conclusions: These data do not provide evidence for an association between SSB consumption and prevalent CAC in adult men and women.

## 1. Introduction

Consumption of sugar-sweetened beverages (SSB), including regular and diet sodas, has increased significantly in recent decades [1]. SSB have a high carbohydrate content and glycemic load, and are associated with weight gain [1,2,3,4], type 2 diabetes mellitus [1,5,6], metabolic syndrome [7], and cardiovascular diseases [8]. Artificially sweetened beverages (e.g., diet soda) have been proposed as an alternative to SSB, but these beverages are also linked with cardiometabolic dysfunction [9,10,11]. 

Previous studies such as the Health Professionals Follow-Up Study and Nurses’ Health Study have suggested that SSB are associated with higher risk of coronary heart disease [12,13]. In a prospective study of 39,786 Japanese men and women aged 40–59 years, soft drink consumption was associated with higher risk of ischemic stroke in women [14]. Singh et al. recently showed a positive association of SSB with mortality from all causes and cardiovascular disease in a large prospective epidemiologic study [15]. Coronary artery calcium (CAC) detected by cardiac computed tomography (cardiac CT) is a subclinical marker of atherosclerosis, which predicts cardiovascular disease [16,17]. The data on the association between SSB consumption and subclinical atherosclerosis are not available. This is especially important as SSB are cheap, readily available, and consumed worldwide. Hence, we sought to test the hypothesis that SSB consumption is associated with higher prevalence of atherosclerotic plaque in coronary arteries in the National Heart, Lung, and Blood Institute Family Heart Study (NHLBI FHS).

## 2. Materials and Methods

### 2.1. Study Population

The NHLBI FHS study was originally designed to identify and evaluate genetic and non-genetic determinants of risk factors for cardiovascular disease, early stage of atherosclerosis, and coronary heart disease (CHD) [18,19]. In short, a total of 5710 subjects participated in baseline clinical examinations from 1993 to 1995. CAC measurements were done with Cardiac CT between 2002 and 2003 in one-third of subjects who were invited to participate in a clinical examination. In addition, the Hypertension Genetic Epidemiology Network Study at the University of Alabama recruited subjects that underwent cardiac CT. Our study population cohort is shown in Figure 1. Briefly, amongst 3337 subjects who had data on cardiac CT, 367 subjects were excluded for prevalent CHD, and 980 subjects did not have data on SSB consumption at the baseline evaluation. The final sample size for current analyses was 1991. An informed consent was provided by each subject, and the study protocol was reviewed and approved by the participating institutions.

### 2.2. Assessment of SSB Consumption

Dietary information was collected through a semi-quantitative food frequency questionnaire that was administered by the staff, the reproducibility and validity of which have been documented before [20,21,22]. Each subject was asked the following question: “In the past year, how often on average did you consume 1 glass of diet soda, regular soda, and fruit-punch/kool-aid?” (Item 63, 64, and 65 in the questionnaire forms). Possible responses were: almost never, 1–3/month, 1/week, 2–4/week, 5–6/week, 1/day, 2–3/day, 4–6/day, and >6/day. Due to sparse data, we combined adjacent categories when creating final exposure categories of almost never, 1–3/month, 1/week, 2–6/week, 1/day, and ≥2/day for stable estimates. We converted the frequency of regular soda, diet soda, and fruit punch/kool aid per month and per week to per day. Finally, we added the frequency of regular soda, diet soda, and fruit punch/kool aid consumption per day to obtain the total number of SSB consumed per day.

### 2.3. Measurement of Calcified Atherosclerotic Plaque in the Coronary Arteries

Cardiac CT examinations were obtained using General Electric Health Systems LightSpeed Plus and LightSpeed Ultra, Siemens Volume Zoom, or Philips MX 8000 machines. The cardiac CT examination details have been reported previously [19]. Cardiac CT images were initially obtained from all sites and the images were then sent electronically to the central CT reading center located at Wake Forest University Health Sciences, Winston Salem, NC. A trained CT analysts identified CAC in the epicardial coronary arteries using Agatston score. Agatston score refers to the amount of calcium based on the area and the density of the calcified plaques as detected by the scan [23]. In this study, the sum of the vessel plaque as averaged from the first and second measurement is reported as the total CAC score.

### 2.4. Other Variables

Smoking cigarettes and consumption of alcohol were obtained during the clinic visit interview. Dietary information was obtained from food frequency questionnaire. Physical activity information during the previous year was evaluated through self-reports. Weight and height were collected from participants wearing scrub suits. Prevalent CHD was determined from a self-reported history of myocardial infarction, percutaneous transluminal coronary angioplasty, or coronary artery bypass graft. All variables used in these analyses were determined during the initial examination (1993–1995). CAC scores were obtained during a follow-up examination (2002–2003).

## 3. Statistical Analysis

CAC was dichotomized into Agatston CAC score above 100 versus less than 100 [19]. We used generalized estimating equations to calculate the prevalence ratios with corresponding 95% confidence intervals for the presence of CAC across categories of SSB consumption. The multivariable model was adjusted for age, sex, field center, body mass index (continuous), smoking (pack years), alcohol intake (current alcohol intake Y/N), physical activity (quartiles of total MET-min/week), and caloric intake (continuous). All analyses were performed using SAS 9.4 statistical software (SAS Institute Inc., Cary, NC, USA). All *p*-values were two tailed and significance level was set at an alpha of 0.05.

## 4. Results

Of the total 1991 subjects, 60% were female and the mean age was 55.0 years. Median Agatston score (IQR) was 0.50 (61.5). Table 1 shows the baseline characteristics by categories of SSB. Higher intake of SSB was more prevalent in younger men, who were more likely to have higher BMI. Subjects with higher consumption of SSB had a lower number of total pack years of smoking and were more likely to consume more calories. SSB consumption was most frequent amongst those who earn more than $75,000/year. 

### 4.1. SSB Consumption

Higher consumption of SSB was not associated with prevalent CAC in a model adjusted for age, sex, BMI, smoking, alcohol, physical activity, field center, and total calories [prevalence ratio (95%CI) of: 1.0 (reference), 1.36 (0.70–2.63), 1.69 (0.93–3.09), 1.21 (0.69–2.12), 1.05 (0.60–1.84), and 1.58 (0.85–2.94) for SSB consumption of almost never, 1–3/month, 1/week, 2–6/week, 1/day, and ≥2/day, respectively (p for linear trend 0.25), as shown in Table 2].

### 4.2. Regular Soda, Diet Soda, and Fruit Punch/Kool Aid Consumption

In a model adjusting for age, sex, BMI, smoking, alcohol, physical activity, field center, and total calories, intake of regular soda was not associated with a higher prevalence of CAC [corresponding prevalence ratios were 1.0 (reference), 0.69 (0.32–1.48), 0.58 (0.30–1.14), 0.72 (0.44–1.18), 0.82 (0.51–1.33), and 1.07 (0.69–1.66), respectively (p for linear trend, 0.44, as shown in Table 2)]. Similarly, higher consumption of diet soda and fruit punch/kool aid was not associated with higher prevalence of CAC in multivariable model with adjustment for age, sex, BMI, smoking, alcohol, physical activity, field center, and total calories [prevalence ratio (95%CI) of: 1.0 (reference), 1.21 (0.73–2.00), 1.44 (0.88–2.36), 0.97 (0.64–1.46), 1.06 (0.58–1.93), and 1.01 (0.59–1.74) for diet soda consumption of almost never, 1–3/month, 1/week, 2–6/week, 1/day, and ≥2/day, respectively (p for linear trend 0.76, as shown in Table 2). Corresponding values for intake of fruit/kool aid were 1.0 (reference), 1.76 (0.51–6.14), 2.12 (0.98–4.59), 1.28 (0.81–2.03), 0.82 (0.50–1.36), and 1.19 (0.80–1.79), respectively (p for linear trend 0.25, as shown in Table 2)].

In a sensitivity analysis, there was no evidence of association between regular soda, diet soda, and fruit punch/kool aid consumption and prevalent CAC when CAC cut points of 0, 50, 150, 200, and 300 were used. 

## 5. Discussion

In our study, we did not find an association between SSB consumption and prevalent CAC in adult subjects free of prevalent CHD after adjustment for cardiovascular risk factors. Using different CAC cut points of 0, 50, 150, 200, and 300 did not change the results.

There has been an increasing research interest in SSB consumption as a possible mediator for cardiovascular disease owing to the high carbohydrate content and glycemic load in SSB. SSB have been shown to be associated with cardiovascular disease and mortality in prospective follow-up studies. In the Nurses’ Health Study, Fung et al. showed that SSB consumption of 1 serving/day and ≥2 servings per day were, respectively associated with 23% (RR 1.23; 9%%CI 1.06–1.43) and 35% (RR 1.35; 95%CI 1.07–1.69) higher risk for incident coronary heart disease [12]. Similarly, de Koning et al., in the Health Professional Follow-up study, showed that participants in the top quartile of SSB consumption had a 20% higher risk (RR 1.20; 95%CI 1.09–1.33) of coronary heart disease than those in the bottom quartile [13]. Eshak et al. in the Japan Public Health Center-based study cohort I showed that participants with ≥1 serving/day of soft drink intake had 83%% higher risk for ischemic stroke in women (HR 1.83; 95%CI 1.22–2.75), compared with participants who never or rarely drank soft drinks [14]. Hence, it is important to know whether SSB are associated with higher prevalence of atherosclerosis in the coronary arteries. To our knowledge, this is the first study to examine whether SSB consumption is associated with prevalent CAC.

In our study, we found no association between SSB consumption and prevalent CAC. It is possible that reverse causation could partially explain a lack of a positive relation in our data. For example, older participants who are obese with co-morbidities could have reduced their SSB intake to become healthy or lose weight. This change in behavior can weaken the positive association between SSB intake and coronary atherosclerosis, which is more common in older participants who are obese with other co-morbidities. In current analyses, we excluded participants with prevalent CHD to mitigate reverse causation.

Mechanisms behind potential harmful relationship between SSB consumption and cardiovascular risk are not completely elucidated. It is possible that the harmful effects of SSB consumption are mediated through physiologic pathways that do not involve atherosclerosis. Sugar-sweetened beverages have been shown to be associated with weight gain [24], type 2 diabetes mellitus [7], hypertension [25], metabolic syndrome [7], and gout [26]. It is unknown whether these effects are results of calories provided from sugars or high fructose corn syrup added in these beverages. Carbonated beverages like Coke and Pepsi have high amounts of fructose corn syrup. Since 1980 s, sucrose has been replaced to a large extent by high fructose corn syrup in carbonated beverages, particularly in North America [27]. High fructose corn syrup is associated with increased cardiovascular risk factors such as postprandial triglycerides, LDL-cholesterol, apolipoprotein-B, and uric acid [28,29]. Fructose-sweetened beverages have also shown to been associated with increase in visceral adipose deposition, non-alcoholic steatohepatosis, and denovo lipogenesis [30], and decreased insulin sensitivity [31]. Aeberli et al., in a randomized control trial, showed that low to moderate consumption of SSB containing high amounts of fructose was associated with an increase in inflammatory biomarkers [32]. Inflammation plays an important role in the pathogenesis of CVD, which could present an additional pathway by which SSB influence the risk of CVD.

Limitations of our study include its cross-sectional design, which limits our ability to determine the temporal relation between SSB consumption and CAC. Information on SSB consumption was based on a self-reported questionnaire, which might have led to an underestimation or overestimation of SSB consumption. Another limitation is the use of a single FFQ at baseline, which limits our ability to capture changes in dietary habits that may have occurred over time. Although the CAC measurement was completed about 7 years after dietary assessment, we did not have baseline CAC measurements to differentiate calcification that was present at baseline from calcification that developed after assessment of beverage consumption. Residual confounding cannot be ruled out and may have occurred due to unmeasured or unaccounted factors. On the other hand, a relatively large sample size, comprehensive dietary questionnaire, availability of information on major cardiovascular risk factors, and usage of different cut points of CAC to define presence of coronary calcification in sensitivity analyses are the major strengths of the study. 

## 6. Conclusions

In conclusion, we found no association between SSB consumption and prevalent CAC in adult men and women. 

## Figures and Tables

**Figure 1 nutrients-13-01775-f001:**
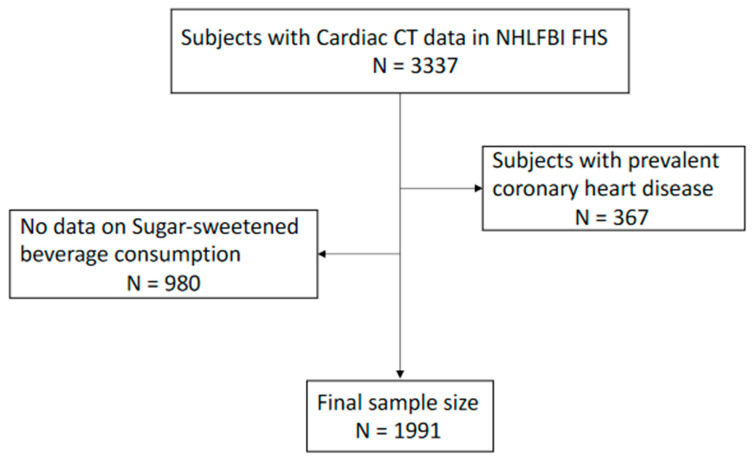
Study population. CT, Computed Tomography; NHLBI FHS, The National Heart, Lung, and Blood Institute Family Heart Study.

**Table 1 nutrients-13-01775-t001:** Characteristics among 1991 participants of the NHLBI Family Heart Study according to sugar-sweetened beverage consumption.

Frequency of Sugar-Sweetened Beverage Consumption (1 Drink)						
	Almost Never (N = 168)	1–3/Month (N = 132)	1/Week (N = 250)	2–6/Week (N = 464)	1/Day (N = 439)	≥2/Day (N = 538)
Age (years)	62.9 ± 12.1	59.7 ± 13.6	58.8 ± 12.2	58.4 ± 12.7	55.8 ± 12.7	52.1 ± 11.9
BMI (kg/m^2^)	27.3 ± 5.1	28.1 ± 5.6	28.1 ± 5.1	28.6 ± 5.7	28.6 ± 5.2	30.1 ± 6.3
Male (%)	23.8	27.3	32.8	41.4	47.2	49.1
Smoking (pack years)	13.2 ± 25.7	8.6 ± 21.1	6.8 ± 15.3	8.9 ± 18.1	7.0 ± 17.0	10.6 ± 19.4
Alcohol (drinks/week)	3.6 ± 6.4	2.6 ± 5.4	2.4 ± 5.0	3.4 ± 10.3	3.3 ± 6.1	4.3 ± 8.8
Hypertension (%)	37.5	34.9	33.6	36.4	38.8	37.4
Diabetes (%)	10.7	7.6	9.2	8.6	10.3	10.4
Income (%)						
<$25,000	19.9	13.6	12.7	12.9	11.0	10.2
$25,000–<$75,000	54.7	56.8	57.0	57.0	53.0	52.5
≥$75,000	15.4	29.6	30.3	30.1	36.0	37.3
Exercise (MET-min/week)	443 (1025)	501 (1028)	436 (802)	443 (775)	376 (872)	358 (886)
Calories (kcal/d)	1385 (663)	1592 (751)	1503 (793)	1561 (696)	1649 (844)	1945 (966)

Age, BMI, and smoking are reported as mean ± SD; exercise and calories are reported as the median (IQR). BMI, body mass index; NHLBI, The National Heart, Lung, and Blood Institute; SD, standard deviation.

**Table 2 nutrients-13-01775-t002:** Prevalence ratios (95%CI) of coronary artery calcification according to sugar-sweetened beverage consumption in 1991 participants from the NHLBI Family Heart Study.

	Cases/n	Age and Sex Adjusted	Model 1 *
Sugar-sweetened beverage (1 glass)			
Almost never	49/168	1	1
1-3/month	34/132	1.08 (0.60–1.96)	1.36 (0.70–2.63)
1/week	70/250	1.36 (0.81–2.29)	1.69 (0.93–3.09)
2-6/week	126/464	1.16 (0.72–1.90)	1.21 (0.69–2.12)
1/day	94/439	0.97 (0.60–1.58)	1.05 (0.60–1.84)
≥2/day	102/538	1.39 (0.82–2.37)	1.58 (0.85–2.94)
p for trend		0.51	0.32
Regular soda (1 glass)			
Almost never	246/858	1	1
1–3/month	62/279	0.55 (0.28–1.06)	0.69 (0.32–1.48)
1/week	53/214	0.56 (0.32–0.98)	0.58 (0.30–1.14)
2–6/week	62/288	0.65 (0.43–0.99)	0.72 (0.44–1.18)
1/day	32/177	0.66 (0.44–0.98)	0.82 (0.51–1.33)
≥2/day	20/175	0.72 (0.49–1.05)	1.07 (0.69–1.66)
p for trend		0.066	0.44
Diet soda (1 glass)			
Almost never	194/878	1	1
1–3/month	46/203	1.48 (0.99–2.21)	1.21 (0.73–2.00)
1/week	36/125	1.45 (0.96–2.18)	1.44 (0.88–2.36)
2–6/week	76/278	1.12 (0.78–1.60)	0.97 (0.64–1.46)
1/day	62/227	1.11 (0.66–1.88)	1.06 (0.58–1.93)
≥2/day	61/280	0.96 (0.61–1.52)	1.01 (0.59–1.74)
p for trend		0.30	0.76
Fruit punch/kool aid (1 glass)			
Almost never	248/900	1	1
1–3/month	89/389	0.87 (0.34–2.25)	1.76 (0.51–6.14)
1/week	49/250	1.06 (0.54–2.08)	2.12 (0.98–4.59)
2–6/week	59/300	0.76 (0.51–1.13)	1.28 (0.81–2.03)
1/day	23/105	0.58 (0.37–0.90)	0.82 (0.50–1.36)
≥2/day	7/47	-- ^#^	-- ^#^
p for trend		0.18	0.25

* Adjusted for age, sex, BMI, smoking, alcohol use, physical activity, total calories and field center. ^#^ Too few cases to calculate the HR with 95%CI. * CI, Confidence Interval; HR, Hazard Ratio.

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
