# Peer review of "Sugar-Sweetened Beverage Consumption and Calcified Atherosclerotic Plaques in the Coronary Arteries: The NHLBI Family Heart Study"

_nutrients, 2021, doi:10.3390/nu13061775_

Round 1

Reviewer 1 Report

- Thanks for this oppurtunity to review your manuscript. 

- Well written manuscript.

- In this study with 1990 participants, authors found no association between SSB and CAC. This is very important question at this time with increased usage of sweetened beverages and increased incidence of atherosclerosis.

  • Major drawback of this study is: SSB consumption data was obtained in 1993-1995 and CAC was measured in 2003-2005. Participants might have changed their behaviour within these 8 years which makes it very hard to come to any conclusion from this data.
  • Recall bias is also integral to this type of studies.

Authors did good job in mentioning these limitations.

Author Response

We want to thank this reviewer for their comments and review.

1. Major drawback of this study is: SSB consumption data was obtained in 1993-1995 and CAC was measured in 2003-2005. Participants might have changed their behavior within these 8 years which makes it very hard to come to any conclusion from this data.

Answer: Yes, this is a limitation of the study as mentioned in the manuscript. We unfortunately do not have CAC measurements at the baseline during the time of food frequency questionnaire collection. We, although believe that it takes few years to develop CAC in the epicardial coronary arteries, and hence there would not have been significant increase in CAC scores from the time of collection of food frequency questionnaire.  Also, given we did sensitivity analysis using different cut off values of CAC with same results, we do not think this would have influenced to a great extent to the results of the study.

2. Recall bias is also integral to this type of studies.

Answer: Yes, recall bias is unfortunately an important bias for any food frequency questionnaire study and not just this particular study.

Reviewer 2 Report

Comments:

The paper entitled “Sugar-sweetened beverage consumption and calcified atherosclerotic plaques in the coronary arteries: the NHLBI Family Heart Study” aims at investigating the role of sugar-sweetened beverage (SSB) intake in the atherosclerosis process. The authors conclude that no association between SSB intake and coronary-artery calcium (CAC) exist. The study is well presented and provides an interesting perspective on a contemporary topic. However, some there are some issues that should be addressed in order to better communicate the message. Please find below my comments:

  1. Although the study design is outlined throughout the paper, the present paper is part of a bigger study and I would suggest to add a flow chart about the study design specifying the number of patients, the patients excluded from the study and the exams/procedures performed
  2. As stated by the author in the limitations paragraph there are many confoundings that could influence the results. One of those is that it is a single snapshot whereas a T0-T1 assessment would have added much information. In this perspective, if at least lab test are available I would add baseline values (T0) and those at the moment of coronary assessment.
  3. Among the presented cardiovascular risk factors the lipidic profile and the familiarity for cardiovascular disease is missing. Probably it would an important information to add.
  4. From the study design I understand that those patients who developed conary cardiovascular disease were excluded from the study. Did the authors think also to give the questionnaire about SSB intake also to those patients and, in this way, to have a comparison group?

Author Response

We want to thank this reviewer for their insightful review.

1. Although the study design is outlined throughout the paper, the present paper is part of a bigger study and I would suggest to add a flow chart about the study design specifying the number of patients, the patients excluded from the study and the exams/procedures performed

Answer: A flowchart of the study population as recommended has been added in figure 1 and mentioned in the manuscript in lines 134-135.

2. As stated by the author in the limitations paragraph there are many confoundings that could influence the results. One of those is that it is a single snapshot whereas a T0-T1 assessment would have added much information. In this perspective, if at least lab test are available I would add baseline values (T0) and those at the moment of coronary assessment.

Answer: We unfortunately do not have any lab values measured during the time of food frequency questionnaire and Cardiac CT measurements. Adding lab values if available would have greatly help to support the hypothesis and the overall results. 

3. Among the presented cardiovascular risk factors the lipidic profile and the familiarity for cardiovascular disease is missing. Probably it would an important information to add.

Answer: Unfortunately, we do not have lab values on lipid profile in these subjects during the time of data collection. Also we had many missing information on family history and hence we did not include this important risk factor as a covariate in the analysis.

4. From the study design I understand that those patients who developed coronary cardiovascular disease were excluded from the study. Did the authors think also to give the questionnaire about SSB intake also to those patients and, in this way, to have a comparison group?

Answer: Only 10% of subjects who had a history of coronary heart disease were excluded from initial cohort. So, we think a direct comparison between the two groups would not had been possible. Collecting food frequency questionnaire and obtaining an association of SSB intake and prevalent coronary heart disease would have been useful. However, obtaining cardiac CT for CAC is not useful in subjects with established coronary heart disease and hence we think a comparison between the two groups would not be justified

Round 2

Reviewer 2 Report

The authors answered to reviewer comments